# UNAAGI: Atom-Level Diffusion for Generating Non-Canonical Amino Acid Substitutions

## Abstract

Proposing beneficial amino acid substitutions, whether for mutational effect prediction or protein engineering, remains a central challenge in structural biology. Recent inverse folding models, trained to reconstruct sequences from structure, have had considerable impact in identifying functional mutations. However, current approaches are constrained to designing sequences composed exclusively of natural amino acids (NAAs). The larger set of non-canonical amino acids (NCAAs), which offer greater chemical diversity, and are frequently used in in-vivo protein engineering, remain largely inaccessible for current variant effect prediction methods.

To address this gap, we introduce **UNAAGI**, a diffusion-based generative model that reconstructs residue identities from atomic-level structure using an E(3)-equivariant framework. By modeling side chains in full atomic detail rather than as discrete tokens, UNAAGI enables the exploration of both canonical and non-canonical amino acid substitutions within a unified generative paradigm. We evaluate our method on experimentally benchmarked mutation effect datasets and demonstrate that it achieves substantially improved performance on NCAA substitutions compared to the current state-of-the-art. Furthermore, our results suggest a shared methodological foundation between protein engineering and structure-based drug design, opening the door for a unified training framework across these domains.

## 1 Introduction

Proteins are linear chains of amino acids that fold into specific three-dimensional structures to perform a wide range of biological functions. Chemically, an amino acid consists of a central $\alpha$-carbon bonded to a hydrogen atom, an amino group ($NH_2$), a carboxyl group (COOH), and a variable side chain denoted as $R$. While the $R$ group can take on diverse chemical forms, only 20 distinct side-chain structures are commonly found in naturally occurring proteins. These are referred to as the genetically encoded, canonical, or Natural Amino Acids (NAAs) (Branden & Tooze, 1991). Several studies suggest that this set of 20 NAAs has been evolutionarily optimized to achieve functional completeness and broad chemical diversity (Ilardo et al., 2015; Doig, 2017; Ilardo et al., 2019). Nevertheless, Non-Canonical Amino Acids (NCAAs) offer side-chain chemistries and functionalities absent from the standard set, such as enhanced metal coordination, tailored electrostatics, or novel catalytic properties. Incorporating NCAAs enables protein engineers to expand the functional repertoire of proteins beyond evolutionary constraints, unlocking new possibilities in synthetic biology and therapeutics (Link et al., 2003; Chin, 2017; Rogers et al., 2018).

In recent years, machine learning – particularly deep learning – has achieved remarkable success in protein research, advancing tasks such as structure prediction, design, and mutational effect estimation (Jumper et al., 2021; Dauparas et al., 2022; Abramson et al., 2024). Variant effect prediction models are often trained in an unsupervised fashion, where amino acid propensities in a protein are predicted conditioned on a sequence (Riesselman et al., 2018) or structural (Dauparas et al., 2022) context. Such procedure generally model amino acids using a discrete distribution, typically limited to the 20 naturally occurring amino acids, based on the propensities observed in our vast databases on evolutionary data. While such models have shown impressive zero-shot predictive performance for various protein properties, the discrete modeling prohibits generalization beyond the naturally occurring amino acids, representing a fundamental limitation in the current modeling paradigm.

A central question is whether inverse-folding models can be generalized beyond the natural amino acids. This question involves two challenges: 1) the vast chemical-space of NCAAs renders categorical modeling inpractical, 2) the predictive capabilities in inverse folding models originates from the evolutionary selection pressure encoded in the millions of naturally occurring proteins in our databases – a source of data not available for non-canonical amino acids.

In this paper, we explore a hypothesis which could provide a potential solution to both challenges: by modeling amino acids generatively in full atomic detail, rather than as a set of discrete tokens, we can obtain some level of generalization from the natural to the non-canonical amino acids. While it is clear that we cannot expect prediction capabilities to extend far out-of-domain, the hope is that it is possible to predict the substitution feasibility for non-canonical amino acids which do not deviate dramatically from their canonical counterparts.

To explore the potential of this idea, we propose a novel generative framework for residue-level sequence design based on atom-level side-chain generation via equivariant molecular diffusion. We term our approach **U**ncanonical **N**ovel **A**mino **A**cid **G**enerative **I**nference (**UNAAGI**). By modeling the atomic coordinates of side chains directly, UNAAGI implicitly infers residue identity and is capable of proposing plausible non-canonical substitutions; a visual illustration of our motivation is shown in Figure 1. We evaluate our model by comparing its generative likelihoods to experimentally measured mutational effect (Rogers et al., 2018; Notin et al., 2023), and find that the performance on non-canonical amino acids is improved considerably over the current state-of-the-art. Importantly, our model maintains consistent performance across both canonical and non-canonical residues.

Specifically, our contributions are as follows:

- We introduce a novel E(3)-equivariant diffusion framework for atom-wise side-chain generation, offering a new approach to residue identity inference.

- We demonstrate the ability of UNAAGI to generalize to the subclass of non-canonical amino acids which are chemically proximal to the natural amino acids.

- We investigate the zero-shot performance of UNAAGI on variant effect prediction, demonstrating meaningful levels of correlation for the identified subclass of non-canonical substitutions, a substantial improvement over the current state of the art.

- On natural amino acids, we quantify the cost of replacing a token-based output with the molecular generation objective of UNAAGI, through a comparison on a subset of ProteinGym.

- We discuss the broader implications and limitations of our approach. In particular, we quantify the cost of replacing the standard token-based output of inverse-folding models with the molecular generation objective of UNAAGI, through a comparison to state of the art zero-shot predictors on ProteinGym for canonical amino acid substitutions. We also discuss the similarities of our UNAAGI to those of structure-based drug design (SBDD), and the potential for a unified approach.

In the following sections, we formalize the generation task at the atom level, describe our E(3)-equivariant diffusion framework UNAAGI, evaluate it on mutational benchmarks, and discuss its implications for protein design and synthetic biology.

## 2 Related Work

Our approach has connections to prior work in molecular diffusion, variant effect prediction and peptide design.

### 2.1 Molecular Diffusion

Hoogeboom et al. (2022) introduced the first framework for E(3)-equivariant diffusion, generating molecular conformations in 3D space via a diffusion process. However, their approach suffered from low generative quality and often produced molecules with disconnected atoms. Subsequent works, including Peng et al. (2023) and Vignac et al. (2023), proposed refinements that incorporated inductive biases from molecular bonding structures, leading to improved sampling quality.

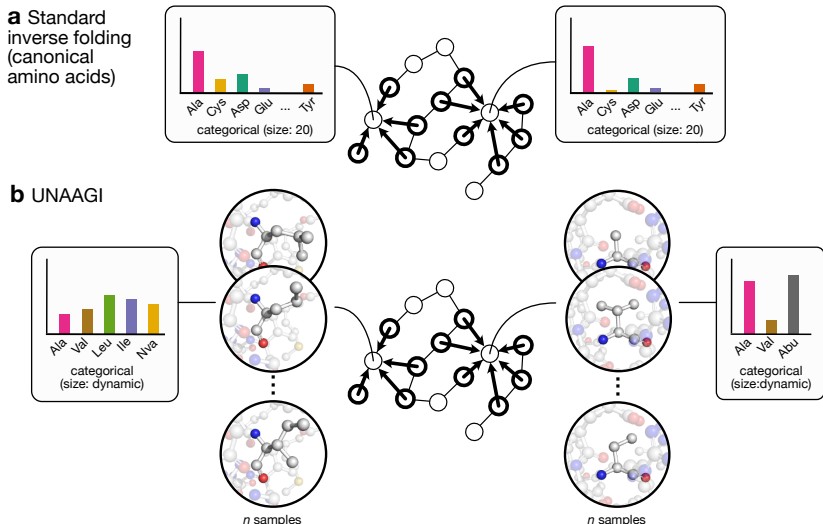

Figure 1: An illustration of the UNAAGI model, compared to a standard inverse folding model. **(a)** Inverse-folding models parameterize the output distribution as a categorical distribution over a fixed size vocabulary **(b)** UNAAGI generates amino acids freely in chemical space; based on a finite number of samples, a histogram is constructed over a dynamical subset of chemical space relevant to the given context, typically including both canonical amino acids (NAA) and non-canonical amino acids (NCAA).

Le et al. (2024) provided a comparative study of several molecular diffusion frameworks, analyzing the impact of different modeling choices on sample quality. Their work identified combinations of design choices that optimize performance for molecular generative tasks.

Beyond unconditional molecular generation, a critical application is the conditional generation of compounds that bind to specific protein targets—known as Structure-Based Drug Design (SBDD). Schneuing et al. (2024) extended the framework of Hoogeboom et al. (2022) to conditional ligand generation in 3D space, while Guan et al. (2023) proposed similar models for target-aware ligand sampling. More recently, Schneuing et al. (2025) advanced this line of research by introducing models capable of sampling ligands with variable atom counts and incorporating protein pocket side-chain conformations during generation.

## 2.2 MUTATIONAL EFFECT PREDICTION

Mutational effect prediction is the task of predicting the impact of amino acid substitutions in a protein on a particular property of interest (e.g. stability, affinity or function). Multiple state-of-the-art protein models—whether sequence-based, structure-based, or hybrids of the two—have demonstrated sensitivity to the mutational landscape of proteins.

Statistical models of multiple sequence alignments remain a strong baseline when many sequences are available for a given protein family. A notable example is the DeepSequence VAE model (Riesselman et al., 2018). Language-based models such as Meier et al. (2021) and Lin et al. (2023) obviate the need for alignments, leveraging conservation and coevolutionary signals learned from massive protein sequence datasets. While sequence-based model can predict mutational effect in many contexts, their performance declines when evolutionary signals are sparse, such as for shallow MSAs, antibodies, or de novo designed peptides.

Structure-based approaches replace the sequence context of a mutation with a structural context. Early models considered the structural environment around one amino acid at a time (Boomsma & Frellsen, 2017; Torng & Altman, 2017), which was later generalized to entire sequences in inverse-folding models (Ingraham et al., 2019; Dauparas et al., 2022; Hsu et al., 2022). Although primarily designed for protein design, these models also show utility in mutational effect prediction Frellsen et al. (2025). Our approach mirrors the early structure-based models by considering only single

amino acids at a time, with the crucial difference that UNAAGI extends mutational effect prediction to include non-canonical amino acids.

## 2.3 Modeling Non-canonical Amino Acid Substitutions

PepINVENT (Geylan et al., 2025) is a transformer-based language model trained on atomic-level peptide representations using CHUCKLES (Siani et al., 1994), which converts amino acid residue tokens into SMILES strings. This enables the model to generate amino acid SMILES for masked positions. The strategy allows for generalization to non-canonical amino acids, but the original study does not validate the sample weights with experimental data. We include this method as a baseline.

NCFlow (Lee & Kim, 2025) and RareFold (Li et al., 2025) are based on AlphaFold3-style architectures. NCFlow adopts the Flow Matching framework and evaluates mutational effect prediction on the same benchmark as ours, allowing for direct comparisons (see Section 4). RareFold introduces new tokens for NCAAs and focuses on predicting protein structures containing NCAA substitutions or designing peptides with NCAAs. Their study reports on successful experimentally validated designed peptides with NCAAs, but the model itself cannot predict affinity scores or mutational effect in silico, making direct comparison on NCAA mutational effect prediction infeasible.

## 3 UNAAGI

We introduce UNAAGI, a diffusion-based model that bridges molecular generation and protein mutational effect prediction by constructing amino acid side chains atom by atom, thereby extending the mutational landscape beyond the natural amino acid space. The objective of the method is to learn patterns in the chemistry of canonical residues, and use these to generalize to chemically proximal non-canonical amino acids. In what follows, we outline the general idea of equivariant graph diffusion, describe the model architecture, training setup, and evaluation protocol, and finally present the baselines used for comparison.

### 3.1 Multi-Modal Diffusion for Molecular Graphs

Diffusion models have emerged as a powerful class of generative models (Sohl-Dickstein et al., 2015; Ho et al., 2020; Song & Ermon, 2020; Song et al., 2021a;b). In the discrete-time setting, these models learn to reverse a Markovian forward process that gradually corrupts data with noise until the samples resemble a tractable prior distribution (typically Gaussian). The generative model is trained to approximate the reverse process, which recovers data via a parameterized denoising chain.

Originally developed for image generation, diffusion models have since been extended to molecular and protein structure generation (Hoogeboom et al., 2022). Let $\mathbf{x}_0 \sim q(\mathbf{x}_0)$ denote a sample from the data distribution. The forward process is defined as

$$q(\mathbf{x}_{1:T} \mid \mathbf{x}_0) = \prod_{t=1}^{T} q(\mathbf{x}_t \mid \mathbf{x}_{t-1}), \tag{1}$$

which incrementally adds noise over $T$ steps until $\mathbf{x}_T$ becomes indistinguishable from Gaussian noise. The reverse process is then modeled as

$$p_\theta(\mathbf{x}_{0:T}) = p(\mathbf{x}_T) \prod_{t=1}^{T} p_\theta(\mathbf{x}_{t-1} \mid \mathbf{x}_t), \tag{2}$$

which gradually denoises $\mathbf{x}_T$ back to a clean sample.

Molecular data is inherently *multi-modal*: atomic coordinates are continuous, while atom types, bond types, and other chemical features are discrete. To handle this, diffusion over discrete variables has been explored through categorical transition kernels—discrete analogues of Gaussian noise in the forward process (Hoogeboom et al., 2021; Austin et al., 2023). Recent molecular diffusion models combine continuous and categorical processes to jointly model the heterogeneous modalities of molecules (Peng et al., 2023; Vignac et al., 2023; Guan et al., 2023; Le et al., 2024).

The forward noise process for continuous and discrete modalities can be expressed as

$$
\begin{aligned}
q(\mathbf{x}_t \mid \mathbf{x}_0) &= \mathcal{N}\big(\mathbf{x}_t \mid \sqrt{\bar{\alpha}_t}\mathbf{x}_0,\ (1 - \bar{\alpha}_t)\mathbf{I}\big), \\
q(\mathbf{c}_t \mid \mathbf{c}_0) &= \mathcal{C}(\mathbf{c}_t \mid \bar{\alpha}_t \mathbf{c}_0 + (1 - \bar{\alpha}_t)\tilde{\mathbf{c}}),
\end{aligned} \tag{3}
$$

where $\bar{\alpha}_t = \prod_{k=1}^{t}(1-\beta_k) \in (0,1)$ is the cumulative noise schedule. For discrete features, $\tilde{\mathbf{c}}$ denotes the categorical prior (e.g., uniform or empirical distribution). Following D3PM (Austin et al., 2023), we adopt a *mask diffusion* scheme, which gradually converts labels into an absorbing state during the forward process.

In our setting, we perturb both atomic coordinates $\mathbf{X}$ and atom-wise categorical features $\mathbf{H}$. The features in $\mathbf{H}$ include atom types, formal charges, hybridization states, aromaticity, ring membership, and atomic degree. Each modality is diffused independently using either Gaussian or categorical noise. This chemically-aware perturbation strategy is inspired by Le et al. (2024), who showed that injecting chemically meaningful noise improves sample quality and stability.

The training objective follows the standard ELBO on the data log-likelihood:

$$
\log p(\mathbf{x}_0) \geq \mathcal{L}_0 + \mathcal{L}_{\text{prior}} + \sum_{t=1}^{T-1} \mathcal{L}_t, \tag{4}
$$

where $\mathcal{L}_0 = \log p(\mathbf{x}_0 \mid \mathbf{x}_1)$ is the reconstruction term and $\mathcal{L}_{\text{prior}} = -\text{KL}(q(\mathbf{x}_T)\|p(\mathbf{x}_T))$ matches the prior. In practice, these terms are often omitted. The main learning signal comes from minimizing the per-timestep KL divergence:

$$
\mathcal{L}_t = \text{KL}[q(\mathbf{x}_{t-1} \mid \mathbf{x}_t, \mathbf{x}_0) \,\|\, p_\theta(\mathbf{x}_{t-1} \mid \mathbf{x}_t)], \tag{5}
$$

which has closed-form solutions under both Gaussian and categorical noise. This is typically optimized by predicting either the clean data $\hat{\mathbf{x}}_0$, the noise $\boldsymbol{\epsilon}$, or categorical logits, depending on the parameterization (Ho et al., 2020; Austin et al., 2023). We adopt the $\hat{\mathbf{x}}_0$ - parameterization to directly predict the clean data. The loss function is mean squared error for atomic coordinates and cross-entropy for other categorical features.

## 3.2 E(3)-Equivariant Graph Neural Network

Molecular structures are inherently three-dimensional and symmetric under Euclidean transformations such as rotation and translation. To respect these symmetries, we incorporate *equivariance* as an inductive bias.

Formally, a function $f : \mathcal{X} \to \mathcal{Y}$ is equivariant to a group $G$ if

$$
f(g \cdot x) = g \cdot f(x), \quad \forall g \in G.
$$

For molecular data, the relevant group is E(3), which includes all 3D rotations and translations. Specifically, a denoising function on atomic coordinates $\mathbf{X} \in \mathbb{R}^{N \times 3}$ should satisfy

$$
f(\mathbf{X}Q + \mathbf{t}) = f(\mathbf{X})Q + \mathbf{t},
$$

for any orthogonal transformation $Q \in \text{O}(3)$ and translation $\mathbf{t} \in \mathbb{R}^3$.

We adopt an E(3)-equivariant Graph Neural Network, closely following the EQGAT-diff model of Le et al. (2024), to learn the score function over atom-level graphs. Each node represents an atom and carries scalar features $h_i$ (e.g., atom type, charge) and vector features $\mathbf{v}_i \in \mathbb{R}^3$, while edges encode bond types and interatomic distances. The details of the training procedure and architectural design are provided in Supplementary A.1.

Message passing is performed on a fully connected graph with attention-based aggregation. The directional information between atoms $i$ and $j$ is encoded as a unit vector

$$
\mathbf{x}_{ji,n} = \frac{\mathbf{x}_j - \mathbf{x}_i}{\|\mathbf{x}_j - \mathbf{x}_i\|},
$$

which is used to construct equivariant vector features. Messages are computed via MLPs over concatenated scalar and edge features, then split into components to update scalar, vector, and positional embeddings in an equivariant manner. This design ensures that all updates preserve E(3)-equivariance and enables joint modeling of geometry, connectivity, and chemistry during denoising.

### 3.3 Graph Topology and Virtual Node Padding

Unlike traditional Structure-Based Drug Design (SBDD), where generative models sample ligands conditioned on a protein pocket, our task focuses on reconstructing amino acid side chains. A key distinction is that each generated side chain is covalently bonded to a fixed protein backbone, imposing local structural constraints around the masked region. To account for this, we reformulate the graph construction. Specifically, the positions and types of backbone atoms for the residue under reconstruction are fixed, anchoring generation to the known structural context.

A further challenge is unified sampling across natural amino acids (NAA) and non-canonical amino acids (NCAA), which vary in atom counts. In standard SBDD models, the atom count must be fixed before diffusion, since the graph topology must be predetermined. We address this using a virtual node strategy inspired by DrugFlow (Schneuing et al., 2025). Unlike empirical sampling schemes (Hoogeboom et al., 2022; Schneuing et al., 2024; Guan et al., 2023) or size-estimation networks (Igashov et al., 2024), this approach is fully end-to-end. This allows sampling from a smoother distribution over amino acid identities across side chains of varying sizes.

Virtual nodes are assigned a special atom type `NOATOM` and are disconnected from all other atoms, with edges labeled `NOBOND`. During training, a random number of virtual nodes $n_{\text{virt}} \sim U(0, N_{\text{max}})$ are added to the side chain. Following Schneuing et al. (2025), these nodes are initialized at the side-chain center of mass. The total number of nodes is capped by a predefined upper bound $N_{\text{max}}$, while the effective side-chain size becomes variable and learnable. At sampling time, the model denoises a graph with $N_{\text{max}}$ nodes. Nodes denoised to the virtual type are removed post hoc, allowing variable-sized side chains to be generated within a unified diffusion framework.

## 4 Experiments

We now describe the experimental procedure for UNAAGI, including training details, evaluation methods, and comparisons against baselines.

### 4.1 Dataset Composition and Preprocessing

We use a dataset of 1,000 protein structures submitted to the Protein Data Bank (PDB) for training. For each structure, we extract every residue along with its local environment, defined as surrounding residues whose center of mass lies within a 10 Å radius of the target residue.

In addition to canonical amino acids, we include non-canonical amino acids (NCAAs) to enhance the model's capacity to generalize to novel chemical environments and to prevent overfitting. For NCAAs, we use datasets from Ilardo et al. (2019) and SwissSidechain (Gfeller et al., 2012), although these consist of NCAAs without associated proteomic context. To capture diverse physical interactions, we also augment the training set with protein–ligand complexes from PDBBind (Wang et al., 2005). Graph construction differs slightly between sources: - For independent NCAAs, the four backbone atoms (C, C, O, N) are fixed during training, and only side-chain atoms are diffused. - For PDBBind, we fix pocket atoms and apply noise to all ligand atoms, following the procedure of SBDD models in Le et al. (2024). We conduct an ablation study on the two supplementary sources of training data to evaluate their individual effects on model performance. Details are provided in Section 4.5.

### 4.2 Evaluation on Deep Mutational Scanning (DMS)

During sampling, UNAAGI generates diverse side-chain conformations given a fixed environment. We interpret the sampling frequencies of side-chain identities and conformations as a proxy for the probability density learned by the diffusion model.

To evaluate predictive power, we compare this learned distribution against experimental Deep Mutational Scanning (DMS) data. Specifically, for each site in the DMS benchmark, we perform 1000 sampling iterations and record frequencies of each sampled amino acid.[1] While larger sample sizes

---

[1]For amino acids that are not sampled, we set their frequency to 1.

would provide smoother frequency estimates, we adopt 1000 iterations as a trade-off between statistical stability and computational feasibility. Sampled side chains are matched to known natural or non-canonical amino acids using graph isomorphism (Weisfeiler & Lehman, 1968) [2]. We repeat the experiment for five times, and report the according error bars to prove stable sampling and statistical significance.

From the sampling frequencies, we estimate the likelihood of each mutant or wild-type residue. We then compute the negative log-likelihood (NLL) for each sampled amino acid, calculating the differential log-likelihood as:

$$\Delta \log \mathcal{L} = -\log P(\text{mutant}) + \log P(\text{wild-type}),$$

where $P(\text{mutant})$ and $P(\text{wild-type})$ denote the estimated probabilities of mutant and wild-type residues, respectively. This log-likelihood difference is correlated with experimental $\Delta\Delta G$ values from the DMS benchmark.

As a sanity check on the model's ability to recover natural amino acids, we randomly selected 25 assays from Notin et al. (2023).[3]

To evaluate performance on NCAAs, we use peptide–protein complexes from Rogers et al. (2018), specifically PDB ID 5LY1 (JMJD2A/KDM4A bound to a macrocyclic peptide CP2) and PDB ID 2ROC (Mcl-1 bound to PUMA). This dataset includes DMS measurements of $\Delta\Delta G$ for substitutions across 20 canonical and 20 non-canonical amino acids. For fairness, we restrict evaluation to substitutions where both the mutant and wild-type residues appear in the sampled data.

### 4.3 RESULTS

#### 4.3.1 PROTEINGYM

We evaluate predictive performance by computing the Spearman correlation between likelihood differences and ground-truth experimental values, following the reporting standard of ProteinGym.

For reference, we also evaluate PepINVENT as a baseline in this setting , although it was not originally tested on DMS benchmarks. Following the repository guidelines, we sample by masking one residue at a time and generating candidate substitutions. This is repeated for all positions in each assay. For peptide–protein complexes, we concatenate peptide and protein sequences as input to the language model. Sampled canonical SMILES are mapped back to amino acids if they are in the benchmark to estimate frequencies.

While UNAAGI does not achieve state-of-the-art performance on any single assay, it still exhibits meaningful correlation with experimental mutational effect across most assays (Figure 2). Importantly, UNAAGI is distinct from all other baselines in Figure 2, as it generates residues in an atom-wise manner rather than sampling from a fixed vocabulary. The experiment is somewhat adversarial, since we for this dataset know a priori that we are restricted to the 20 amino acids. It is encouraging that the models produces reasonable performance despite the more expressive output distribution.

For a fair comparison, when UNAAGI is evaluated against its closest methodological baseline, PepINVENT, which represents sequences and samples residues atom by atom, it consistently outperforms PepINVENT on all assays. Notably, PepINVENT shows little to no correlation on most assays, whereas UNAAGI achieves substantial predictive performance.

Since both models operate in an atom-wise sampling regime, they must demonstrate sufficient capacity to reconstruct wild-type residues. To quantify this, we define the *wild-type coverage rate*—the frequency with which the wild-type residue is successfully sampled across all positions in an assay. Results are shown in Table 1, where UNAAGI achieves higher coverage rate by a large margin, while PepINVENT rarely recovers the wild-type residue. We provide the wild-type coverage rate for each assay in the Supplementary Materials (see Supplementary A.2 Fig. 6).

---

[2]We used the NetworkX 3.5 implementation of `is_isomorphic`, which is based on the VF2 algorithm (Cordella et al., 2001).

[3]Note that the computational cost of UNAAGI scales linearly with protein size, since only local structural environments are considered. Scaling to larger proteins is therefore not fundamentally problematic; however, given a limited computational budget, we opted not to run experiments on the full ProteinGym benchmark and instead used a randomly selected subset

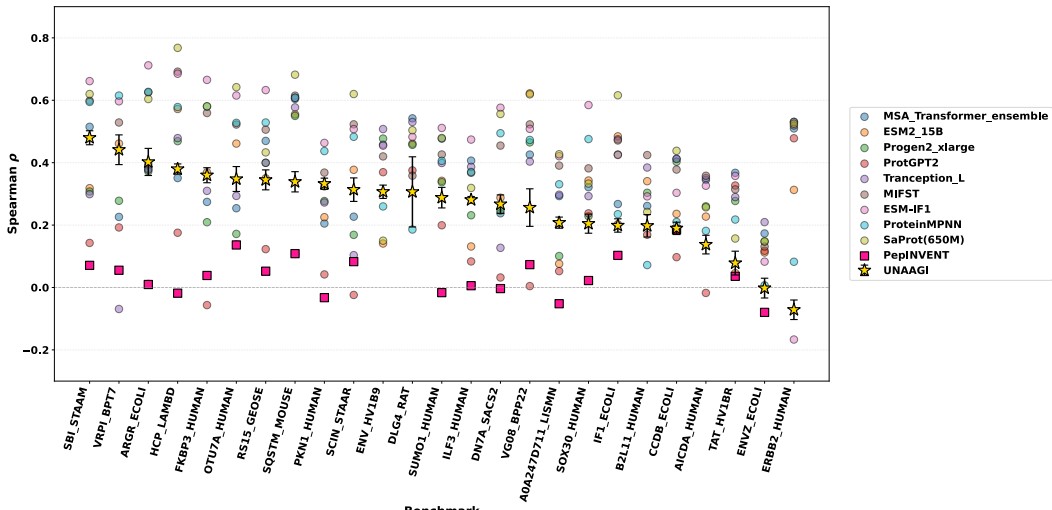

Figure 2: Results of UNAAGI on 25 ProteinGym assays, evaluated using Spearman correlation against experimental DMS measurements.

| Model | Wild-Type Coverage Rate |
|---|---|
| UNAAGI (ours) | 0.9368 |
| PepINVENT | 0.2365 |

Table 1: Comparison of wild-type coverage rates for UNAAGI and PepINVENT, averaged across all benchmarks, including those introduced in Section 4.4.

## 4.4 DMS FOR NCAA SUBSTITUTIONS

We further evaluate UNAAGI on a DMS benchmark containing substitutions to non-canonical amino acids (NCAAs). For comparison, we include both PepINVENT and NCFlow, as NCFlow was evaluated on the same benchmark in its original work.

As shown in Figure 3, UNAAGI achieves consistent performance across both canonical and non-canonical substitution benchmarks. The correlations observed on NCAA substitutions are comparable to those obtained on canonical amino acids, demonstrating that UNAAGI can generalize beyond the natural amino acid space. In contrast, PepINVENT shows limited signal on NCAA substitutions, and NCFlow performs poorly across the NCAA benchmark, with no or negative correlations regardless of the affinity prediction module used. These results indicate that UNAAGI is the first diffusion-based approach to provide reliable predictive power on NCAA mutational effect benchmarks.

## 4.5 ABLATION STUDIES

To assess the influence of each supplementary training data source on UNAAGI, we perform a series of ablation studies in which we train separate models with selected components of the training data removed. Specifically, we remove (i) the PDBBind data, (ii) the independent NCAA data, and (iii) both simultaneously, resulting in three ablated models with different levels of data reduction, as summarized in Table 2. UNAAGI (Ablation: PDBBind) and UNAAGI (Ablation: NCAA) denote models trained without PDBBind and without the independent NCAA dataset, respectively. We evaluate the effect of each ablation by measuring the average Spearman correlation across ProteinGym assays.

Interestingly, we find that removing PDBBind has the smallest impact on UNAAGI's performance, whereas removing the independent NCAA data leads to a substantial drop in correlation.

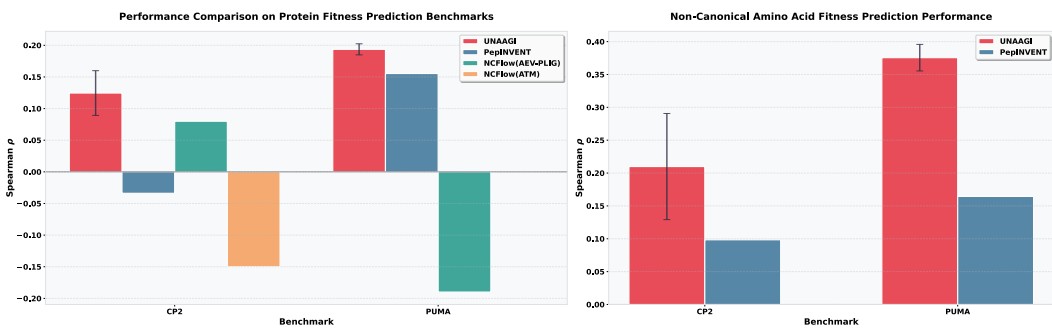

Figure 3: Results of UNAAGI on NCAA DMS benchmarks. NCFlow (ATM) was reported in the original work to fail entirely on the PUMA assay. **Left**: Spearman correlations compared to PepIN-VENT and NCFlow variants with different affinity modules. **Right**: Performance on NCAA substitutions specifically.

Table 2: Ablation studies over the training dataset.

| Model | Spearman $\rho$ |
| --- | --- |
| UNAAGI | 0.2509 |
| UNAAGI (Ablation: PDBbind) | 0.2413 |
| UNAAGI (Ablation: All) | 0.2184 |
| UNAAGI (Ablation: NCAA) | 0.1404 |

## 4.6 ANALYSIS OF SAMPLED NON-CANONICAL AMINO ACIDS

To further investigate the behavior of UNAAGI, we manually inspected a set of sampled amino acids, as shown in Figure 4. We observed two main phenomena:

1. **Chemical diversity.** UNAAGI generates amino acids spanning a broad range of chemistries, including hydrophobic, polar uncharged, positively charged, and negatively charged residues.

2. **Limited structural deviation from natural amino acids.** Although UNAAGI produces non-canonical amino acids (NCAAs), these samples tend to remain structurally close to the 20 natural amino acids. The model often generates NCAAs that resemble existing residues or appear to interpolate between them, rather than introducing more radical structural variations.

In the benchmark described in Section 4.4, we find that UNAAGI consistently recovers only a subset of the NCAAs included in the benchmark—specifically **Nle**, **Nva**, **Abu**, and **tBu**, all of which are hydrophobic. However, UNAAGI does not reliably sample other classes of NCAAs, such as those containing *N*-methyl substitutions or more complex ring systems. A more detailed analysis of NCAA coverage is provided in the Supplementary Material A.3.

## 5 CONCLUSION

In this paper, we introduced UNAAGI, a molecular diffusion model that generates amino acid side chains in an atom-wise manner and can sample across both canonical and non-canonical amino acids. We evaluated UNAAGI on Deep Mutational Scanning (DMS) benchmarks and found that it achieves meaningful correlations in variant effect prediction. Crucially, this correlation extends to benchmarks containing NCAA substitutions, making UNAAGI the first machine learning method to provide measurable signal on this problem.

Methodologically, UNAAGI follows the molecular diffusion paradigm developed for Structure-Based Drug Discovery (SBDD), learning the density of feasible chemical combinations in protein

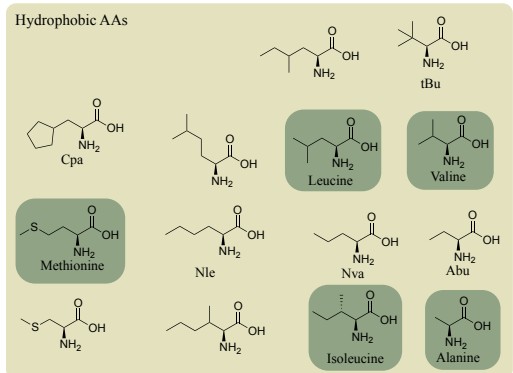

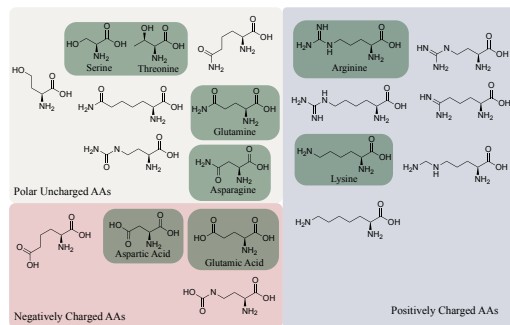

(a) Sampled hydrophobic amino acids from UN-AAGI, including both natural and non-canonical examples (not an exhaustive set).

(b) Sampled polar and charged amino acids from UN-AAGI, including both natural and non-canonical examples (not an exhaustive set).

Figure 4: Qualitative inspection of sampled NCAAs.

structures. This suggests a promising connection between SBDD and variant effect prediction, as both are governed by the same non-covalent interaction principles. Building on this insight, future work could integrate additional SBDD techniques—such as pharmacophore-based conditioning (Peng et al., 2025) or flexible side-chain modeling (Schneuing et al., 2025)—to further improve variant effect prediction.

We acknowledge that in its current form, UNAAGI has important limitations. It remains far from solving the problem of NCAA variant effect prediction and cannot yet propose high-fitness NCAA substitutions efficiently. Although UNAAGI achieves substantial correlations on the NCAA mutants it successfully samples, coverage remains limited to a small subset of the 20 NCAAs in the benchmark (as shown in Section 4.6). This limitation is largely driven by the scarcity of relevant NCAA data. Moreover, UNAAGI tends to interpolate between canonical-like structures, while sampling chemically distinct NCAAs remains an out-of-distribution challenge under the current setup.

We also observe substantial variability in predictive performance across different assays, a phenomenon shared by all methods. This highlights the intrinsic complexity of the variant effect prediction problem: no single model performs universally well, and outcomes depend strongly on model design, training strategy, and data availability.

Looking forward, UNAAGI opens several avenues for exploration. A natural next step is scaling the model, both in parameter count (beyond the current 3.6M) and in training data (from the 1,000 PDB structures used here toward the full database). Incorporating protein structures with NCAA substitutions—although sparse in the PDB—could further expand coverage of the NCAA chemical space. Finally, applying guidance or auxiliary correctors may enhance the generation of chemically diverse NCAAs and enable biasing toward desirable properties, such as synthetic accessibility.

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

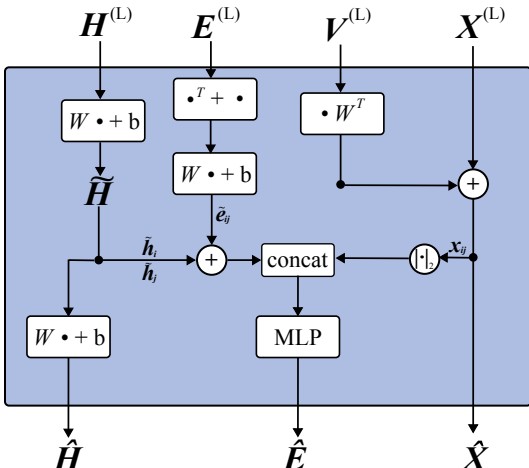

Figure 5: Backbone architecture of UNAAGI, closely following EQGAT-diff (Le et al., 2024).

Boris Weisfeiler and A. A. Lehman. A Reduction of a Graph to a Canonical Form and an Algebra Arising During This Reduction. *Nauchno-Technicheskaya Informatsia*, Ser. 2(N9):12–16, 1968.

# A   APPENDIX

## A.1   TRAINNG DETAILS

### A.1.1   ARCHITECTURE DETAILS

Our model architecture and hyperparameter choices closely follow EQGAT-diff Le et al. (2024). The message passing function at layer $l$ is defined as

$$\mathbf{m}_{ji}^{(l)} = \mathrm{MLP}\Big(\big[\mathbf{h}_j^{(l)} \,;\, \mathbf{h}_i^{(l)} \,;\, \mathbf{W}_{e_0}^{(l)}\mathbf{e}_{ji}^{(l)} \,;\, d_{ji}^{(l)} \,;\, d_j^{(l)} \,;\, d_i^{(l)} \,;\, \mathbf{p}_j^{(l)} \cdot \mathbf{p}_i^{(l)}\big]\Big)$$

where ";" denotes concatenation and the MLP is a 2-layer perceptron. This message embedding further split to sub-embeddings,

$$\mathbf{m}_{ji}^{(l)} = \big(\mathbf{a}_{ji}^{(l)},\, \mathbf{b}_{ji}^{(l)},\, \mathbf{c}_{ji}^{(l)},\, \mathbf{d}_{ji}^{(l)},\, \mathbf{s}_{ji}^{(l)}\big)^{\top} \in \mathbb{R}^K$$

The node, edge, vector, and coordinate updates are then computed as

$$\mathbf{h}_i^{(l+1)} = \mathbf{h}_i^{(l)} + \sum_j \frac{\exp(a_{ji}^{(l)})}{\sum_{j'} \exp(a_{j'i}^{(l)})} \mathbf{W}_h^{(l)}\mathbf{h}_j^{(l)} \qquad \text{and} \qquad \mathbf{e}_{ji}^{(l+1)} = \mathbf{W}_{e_1}^{(l)}\sigma(\mathbf{e}_{ji}^{(l)} + \mathbf{d}_{ji}^{(l)}),$$

$$\mathbf{v}_i^{(l+1)} = \mathbf{v}_i^{(l)} + \frac{1}{N}\sum_j \mathbf{x}_{ji,n} \otimes \mathbf{b}_{ji}^{(l)} + \big(\mathbf{1} \otimes \mathbf{c}_{ji}^{(l)}\big) \odot \mathbf{v}_j^{(l+1)}\,\mathbf{W}_v^{(l)},$$

$$\mathbf{x}_i^{(l+1)} = \mathbf{x}_i^{(l)} + \frac{1}{N}\sum_j s_{ji}^{(l)}\,\mathbf{x}_{ji,n}.$$

Where $\mathbf{1} = (1, 1, 1)^{\top}$ and $\sigma$ is the SiLU activation function. After L layers of message passing, the resulting embeddings are further transformed following Figure 5 to obtain the final representations. Categorical atomic features including hybridization state, formal charges, atom types, aromaticity, ring membership and atomic degree are concatenated to form $\hat{H}$. We leverage the final-layer embeddings $\hat{x}_0 = (\hat{\mathbf{X}}, \hat{\mathbf{H}}, \hat{\mathbf{E}})$ to perform the denoising predictions.

### A.1.2 HYPERPARAMETERS

In all experiments, we use 500 diffusion timesteps for both training and sampling. Scalar feature dimensions are set to 256, vector feature dimensions to 64, and edge feature dimensions to 32. We use 7 layers of message passing, resulting in a total of approximately 3.6M trainable parameters.

For optimization, we use Adam with a learning rate of $5 \cdot 10^{-4}$, and weight decay of $1 \cdot 10^{-4}$. Gradients are clipped at a maximum norm of 10. We maintain an exponential moving average (EMA) of the model weights with decay 0.9999. The batch size during training is set to 8.

### A.2 DETAIL VISUALIZATION OF WILD-TYPE COVERAGE RATE

We show the wild-type coverage per assay in Figure 6. As illustrated there, UNAAGI consistently outperforms PepINVENT across all benchmarks by a large margin.

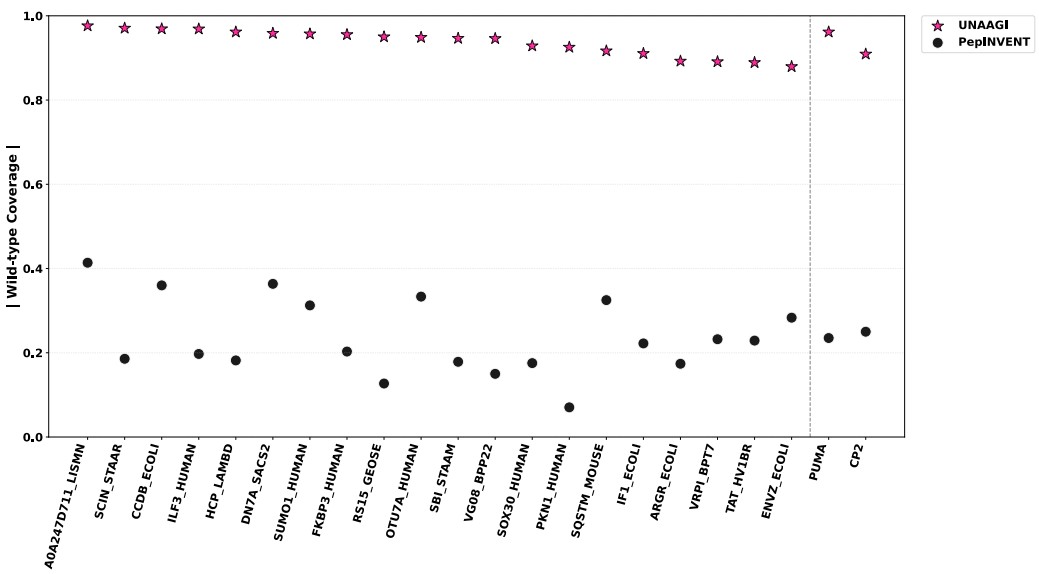

Figure 6: Wild-type coverage rate of UNAAGI and PepINVENT across benchmark assays.

### A.3 COMPREHENSIVE ANALYSIS OF NCAA COVERAGE IN ROGERS ET AL. (2018)

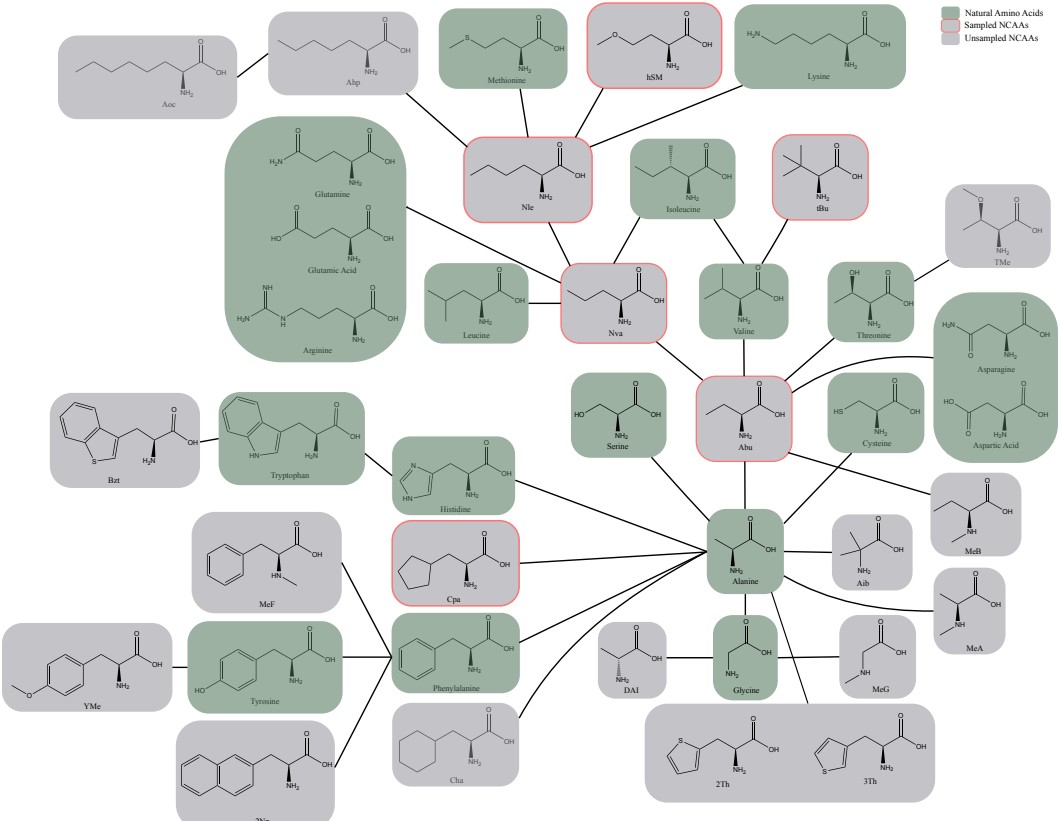

Figure 7: The relationship between the natural amino acids and the NCAAs from the benchmark of Rogers et al. (2018). Natural amino acids are shown with a green background, NCAAs with a gray background, and NCAAs sampled by UNAAGI are highlighted with a red outline. Edges indicate that two amino acids are neighbors, defined as being one graph edit apart.

## A.4 THE USE OF LARGE LANGUAGE MODELS (LLMS)

In accordance with ICLR requirements, we disclose the role of LLMs in preparing this work. Large language models were not used for any experimental design, data analysis, or implementation. Their use was limited to assisting with minor language editing, such as correcting grammar and polishing phrasing.

