# OpenReview forum: "UNAAGI: Atom-Level Diffusion for Generating Non-Canonical Amino Acid Substitutions"
_ICLR.cc/2026/Conference — Submitted to ICLR 2026_

### Official Review · Reviewer_4yJM · 2025-10-18

**Soundness:** 2
**Presentation:** 1
**Contribution:** 2
**Rating:** 2
**Confidence:** 4

**Summary:**

The paper introduces UNAAGI, an **$E(3)$-equivariant diffusion-based generative model** designed for the reconstruction of residue identities from atomic-level structure. A notable feature is its ability to explore both **canonical and non-canonical amino acid substitutions** within a unified generative paradigm. The method is evaluated on several experimentally benchmarked mutation effect datasets. Overall, the work addresses an important problem in protein design, but improvements in clarity, novelty exposition, and experimental validation are necessary.

**Strengths:**

1.  **Novel Generative Task**: The inclusion of non-canonical amino acid substitution as a generative task is a significant and novel contribution. This method expands the scope of computational protein design beyond the traditional 20 canonical residues, opening promising new avenues for exploring protein functionality, stability, and therapeutic applications using synthetic or engineered residues.

**Weaknesses:**

1.  **Questionable Methodological Novelty**: The proposed UNAAGI model appears to be a combination of existing methodologies, and the manuscript does not sufficiently elucidate the key, specific technical innovations that distinguish this model from its predecessors in the context of the presented tasks.

2.  **Insufficient Experimental Baselines**: The experimental evaluation is limited by a narrow selection of baseline methods. it omits comparisons with several highly relevant state-of-the-art models for side-chain prediction and mutation effect prediction, which often handle atomic-level or torsional representations of side-chains.

3.  **Lack of Clarity in Method Description**: The paper suffers from a lack of sufficient detail regarding the architectural specifics and the training regimen of the UNAAGI model. This absence of clarity. The authors should improve the methodological description.

**Questions:**

### Novelty and Methodological Clarification

1.  Could the authors explicitly clarify the specific, novel methodological contributions of UNAAGI? As the current description suggests a combination of pre-existing components.

2.  The statement regarding the novelty of "modeling sidechain in full atomic detail rather than as discrete tokens" requires refinement. Previous works, such as those employing methods like FAMPNN [1] or torsional diffusion models [2], have also modeled side-chains using atomic coordinates or torsion angles. Could the authors provide a more precise explanation of how UNAAGI's approach to atomic-level side-chain modeling differs fundamentally or offers an advantage over these existing methods?

3.  Could the authors include and compare their work with contemporary side-chain modeling approaches specifically tailored for protein mutation effect prediction in the Related Work section, such as those referenced in [3] and [4].

4.  In the Method section, clarification is needed regarding the processing of coordinates for masked tokens. Are these coordinates zero-padded, initialized to a specific value, or handled in an alternative manner during the forward and reverse diffusion processes?

### Experimental Design and Evaluation

5.  The rationale behind several choices in the Dataset Curation section must be clarified:
    * Why was the PDB subset restricted to only 1000 proteins? Is this sufficient for robust model training, particularly for a diffusion model?
    * What specific procedures were implemented to address and mitigate potential data overlap between the training, testing, and external evaluation sets, particularly concerning the NCAAs and PDBBind datasets?

6.  Regarding the evaluation on DMS datasets, why was the model's performance estimated using a sample frequency derived from 100 iterations rather than utilizing the likelihood or score function directly provided by the diffusion model? Clarification on the statistical justification for this sampling approach is required.

7.  Could the authors justify the decision to only evaluate on assays containing fewer than 100 residues? This size constraint may limit the generalizability of the reported performance to larger or more complex proteins.

8.  In the Comparison on the ProteinGYM dataset, the evaluation is incomplete. The authors must include a comparison with recent high-performing methods on this benchmark, such as SaProt [5] and other relevant models, to provide an up-to-date and authoritative performance assessment.




References:

[1] Widatalla, T., Shuai, R.W., Hie, B. and Huang, P., Sidechain conditioning and modeling for full-atom protein sequence design with FAMPNN. In Forty-second International Conference on Machine Learning.

[2] Zhang, Y., Zhang, Z., Zhong, B., Misra, S. and Tang, J., 2023. Diffpack: A torsional diffusion model for autoregressive protein side-chain packing. Advances in Neural Information Processing Systems, 36, pp.48150-48172.

[3] Liu, S., Zhu, T., Ren, M., Yu, C., Bu, D. and Zhang, H., 2023. Predicting mutational effects on protein-protein binding via a side-chain diffusion probabilistic model. Advances in Neural Information Processing Systems, 36, pp.48994-49005.

[4] Luo, S., Su, Y., Wu, Z., Su, C., Peng, J. and Ma, J., Rotamer Density Estimator is an Unsupervised Learner of the Effect of Mutations on Protein-Protein Interaction. In The Eleventh International Conference on Learning Representations.

[5] Su, J., Han, C., Zhou, Y., Shan, J., Zhou, X. and Yuan, F., SaProt: Protein Language Modeling with Structure-aware Vocabulary. In The Twelfth International Conference on Learning Representations.

---

> ### Author Response · Authors · 2025-12-02
> **Response to Reviewer 3 (4yJM)**
>
> ## Questions
>
> ## 1–3: Clarification of Novelty and Relation to Prior Work
>
> We thank the reviewer for their review of our paper. We believe there may be some misunderstanding regarding how UNAAGI fundamentally differs from the models cited by the reviewer. In particular, the reviewer states that our method is of "Questionable Methodological Novelty", and that we include "Insufficient Experimental Baselines", but the baselines that the reviewer refers to solve a different problem than our method. We attempt to clarify these points below, and have also rewritten the introduction of the paper to make the objective of the paper clearer.
>
> ### 1. Core contribution of UNAAGI.
>
> UNAAGI is a **fully atomic, continuous diffusion model that generates both the positions and chemical identities of every atom** in an amino acid side chain. Importantly, however, it is designed with the purpose of sampling non-canonical amino acids, i.e. extending beyond the 20 naturally occurring amino acids.
>
> It does not rely on evolutionary information (no MSAs) nor on discrete amino acid labels. Despite this minimal conditioning, UNAAGI achieves meaningful correlations on mutational effect prediction and, importantly, can **naturally extend to NCAAs** through continuous sampling in chemical space.
>
> ### 2. Why prior works (FAMPNN, DiffPack, etc.) do not cover our setting.
>
> The works cited by the reviewer address side-chain packing, where the amino acid identity is known a priori. These models predict conformations or rotamers conditioned on a fixed residue type. Even FAMPNN—which predicts residue identity before packing—ultimately assumes the identity is selected from a **discrete vocabulary**, and its structural prediction is restricted to canonical residue topologies.
> In contrast, UNAAGI jointly generates (i) atom coordinates and (ii) atom types (C, N, O, S, etc.), allowing the creation of **variable-sized, variable-topology side chains**, which is essential for generalization to NCAAs. None of the referenced models support this capability.
>
> ### 3. Why RDE and related methods differ conceptually.
>
> The methods in [3] and [4] connect side-chain conformational modeling to mutational effect prediction, but they require the amino acid identity as input and operate on rotamer distributions. UNAAGI, by contrast, **predicts atomic structures without residue identity**, enabling zero-shot sampling of NCAAs and capturing mutational effects through a generative density rather than rotamer classification.
>
> ### 4. Model Details and Architecture
>
> We have expanded the description of our architecture, hyperparameters, and training procedures in the revision.
> The technical details of the virtual-node mechanism are already described in Section 3.3: virtual nodes are initialized at the center of mass during training, and during sampling the model learns to denoise certain atoms into virtual nodes, enabling variable atom counts.
>
> ### 5. Dataset Size and Potential Overlap
>
> We restricted training to ~1,000 PDB structures due to computational constraints, but note that residue-wise reconstruction yields ~500,000 training examples.
> Since UNAAGI is trained **unsupervised** (it does not use mutational supervision) and ProteinGym prediction is fully zero-shot, potential overlap does not introduce label leakage.
>
> ### 6. Use of Sampling Frequency
>
> We use sampling frequency as an approximation to the conditional density because the likelihood requires structural models of each mutant, which are unavailable unless generated explicitly. UNAAGI may sample multiple distinct conformations for the same residue type, making likelihood estimation non-trivial without extensive diffusion sampling.
>
> ### 7. Protein Length in Evaluation
>
> We evaluated on 20 assays with up to 100 residues due to the computational cost of diffusion sampling. To address the reviewer’s concern, we additionally evaluated five larger assays. The model maintains consistent performance on these larger proteins, indicating that our conclusions are not an artifact of sequence length.
>
> ### 8. Additional Baselines
>
> As requested, we added **SaProt** to the baselines in Figure 1 alongside ProteinMPNN, ESM-IF1, and MIF-ST to provide more comprehensive comparisons.
>
> ## Summary
>
> We hope the above clarifications help address the reviewer’s questions and resolve the points that may have led to confusion. Some of the concerns raised seem related to misunderstandings or to details already explained in the manuscript (e.g., Section 3.3), and we have reiterated these points here for clarity. We appreciate the reviewer’s feedback and would be grateful if they could re-examine the manuscript in light of these clarifications.

---

### Official Review · Reviewer_GzCZ · 2025-10-30

**Soundness:** 2
**Presentation:** 2
**Contribution:** 2
**Rating:** 4
**Confidence:** 4

**Summary:**

This paper introduces UNAAGI, a model based on E(3)-equivariant diffusion that predicts the impact of amino acid substitutions by generating side chains at the atomic level. Its core innovation lies in the ability of the model to operate on a continuous molecular structure representation, thereby unifying the prediction space for natural amino acids (NAA) and non-natural amino acids (NCAA). This work addresses an important gap in computational protein engineering, where most state-of-the-art models are limited to the 20 natural amino acids. However, these advantages are significantly compromised by a lack of experimental rigor, particularly small training datasets, limited and opaque coverage of benchmark NCAAs, and the need for more comprehensive comparative analyses.

**Strengths:**

1. Shifting from discrete symbols (tokens) prediction to continuous, atom-by-atom side-chain generation is an interesting work. Traditional inverse folding models (such as ProteinMPNN and ESM-IF1) are fundamentally limited by their fixed output vocabulary. UNAAGI cleverly circumvents this constraint by modeling the underlying atomic coordinates directly.
2. Choosing an E(3)-equivariant Graph Neural Network architecture is a methodologically sound and principled choice. Since molecular data is inherently three-dimensional, respecting Euclidean symmetries (rotations, translations) is a crucial inductive bias that helps build more data-efficient and robust models.
3. The model in this paper showed a meaningful positive correlation on the NCAA DMS benchmark test.

**Weaknesses:**

1. The experimental evaluation strategy has significant weaknesses, which undermine the credibility of UNAAGI's performance conclusions on natural amino acids. The practice of subset selection based on protein length lacks sufficient justification and may introduce systematic bias. This leads to selection bias, as small proteins are more likely to consist of globular proteins with a single domain, whose properties are mainly determined by local interactions. UNAAGI is a model that relies on local environments. Therefore, this carefully selected subset may have precisely amplified the model's strengths, making its performance appear better than its actual performance on more complex large proteins or multi-domain proteins.
2. The baseline comparison in the paper is unreasonable. UNAAGI is itself a structure-based model, so structure-based inverse folding models (such as ProteinMPNN, ESM-IF1) and hybrid models (such as MIF-ST, SaProt) should be used as baselines for comparison, whereas comparing with mainly sequence-based models would greatly reduce the informativeness and persuasiveness. From Figure 2, it can be seen that the Spearman correlation coefficients of UNAAGI are mostly between 0.1 and 0.4, which is likely much lower than current SOTA structure models.
3. Data scarcity is a major challenge in NCAA modeling. The model was trained using only a small number of PDB structures, which limits the generalization ability of the model. The success on a few NCAAs may be because these NCAAs are structurally simple or chemically similar to natural amino acids (i.e., they are "interpolated" results). If the model performs poorly on NCAAs with more distinctive chemical properties, then the paper's claim about the model's "generalization ability to NCAAs" is unfounded.
4. The paper uses the results of NAA on ProteinGym as a reasonableness check and points out that "it has not fully reached the state-of-the-art level." However, as can be seen from Figure 2, the average Spearman correlation coefficient of ProteinGym replacing the SOTA models on benchmark tests can reach about 0.70 or even higher. In contrast, the performance of UNAAGI is below 0.4 on many test sets, showing significant inadequacy. The authors have not provided a reasonable analysis.
5. Lack of Argument: whether a model jointly trained on SBDD and side-chain generation tasks can learn a more general "protein-ligand interaction grammar," is lacking theoretical support and argumentation.
6. Lack of a section on qualitative analysis, presenting the generated 3D structure, and discussing its chemical feasibility. Adding some qualitative examples can improve the quality of the paper. For instance, an NCAA side chain successfully generated in a protein environment with a high adaptability score; a side chain that failed to generate or is poorly positioned; a novel NCAA generated by the model that has not been seen in the training set; etc.. This will provide more intuitive information for understanding the model's behavior.

**Questions:**

1. 100 samples are clearly insufficient to estimate the probability density of a vast chemical space. Could the authors provide an analysis of sampling stability (e.g., by repeating the 100-sample
experiment multiple times and observing the variance in the correlation score)?
2. The paper admits that the coverage of NCAA is "limited to a small subset". Could the authors quantify which specific NCAA types the model successfully sampled and the proportion of these in the 20 types in the NCAA benchmark (e.g., CP2 and PUMA)? Was the high correlation in Figure 4 calculated based only on this small subset? This is crucial for evaluating the model’s generalization ability and avoiding selection bias.
3. The mixed strategy of training data (PDB, PDBBind, isolated NCAAs) is confusing. Can the authors provide ablation experiments? Specifically: (a) How does the model perform when trained only on 1000 PDB structures (only NAA)? (b) What is the contribution of the PDBBind ligand data and isolated
NCAA data?
4. Regarding the trade-off in NAA performance: The model’s performance on NAA (ProteinGym) is far below SOTA. Is this a fundamental cost for generalizing NCAA, or is it merely due to the model/data scale being too small (3.6M parameters, 1000 PDB structures)?

---

> ### Author Response · Authors · 2025-12-03
> **Response to Reviewer 2 (GzCZ) - 1**
>
> ## Weakness
>
> ### Protein length and potential bias.
>
> Thank you for highlighting this point. We agree it is important to ensure that performance is not confined to small proteins. We therefore evaluated UNAAGI on additional assays containing longer sequences. The new results confirm that the model maintains its predictive ability on these larger proteins as well.
>
> ### Baseline completeness.
>
> We were somewhat surprised by the statement that the "baseline comparison in the paper is unreasonable". Our original manuscript contained comparisons to two other structure-based models (ProteinMPNN and ESM-IF) on the natural amino acids (where these models are applicable). As the reviewer suggests, we have now added MIFST and SaProt.
>
> ### Data-scarcity in NCAA modelling.
>
> We believe there is some misalignment between the expectations of the reviewers and the intended contribution of our work. We agree with the reviewer that data scarcity is a critical problem in NCAA modelling. In response to this challenge, the goal of our work was to investigate whether we could transfer information from the natural occurring amino acids to the NCAAs that are chemically similar. We do not disagree with the reviewer's observation that "The success on a few NCAAs may be because these NCAAs are structurally simple or chemically similar to natural amino acids (i.e., they are "interpolated" results).", but we consider this a positive result. Our goal is not to generalize across the entire chemically diverse NCAA space (which would require much more data). Rather, we aim to demonstrate that a model trained purely on NAA-containing PDB structures, using atomic-level continuous diffusion, can successfully sample NCAAs that lie close to the natural amino acid manifold. Our results confirm such interpolation behavior and provide an initial step toward continuous chemical-space modeling. We acknowledge that this goal was not phrased clearly enough in the original submission. We have rewritten the introduction to make this point clearer, and have included a figure to illustrate which NCAAs we can expect to generalize to (Figure 4 and Figure 7; also see also the response to reviewer joyj)
>
> ### ProteinGym underperformance
>
> We acknowledge that UNAAGI does not reach state-of-the-art performance on ProteinGym. It is important to stress that the ProteinGym benchmark was included as a sanity check; and that performance on the natural amino acids is not the objective of our model. We include the task to verify the drop in performance we see when switching from a categorical task to a generative task. We ask reviewers to consider that UNAAGI tackles a significantly more difficult generative problem: every atom in the side chain must be generated with correct coordinates and correct types, and the structure must then pass exact graph-isomorphism matching to be recognized as a valid amino acid.
>
> This is substantially more challenging than selecting a token from a fixed vocabulary of size 20, as done by discrete inverse-folding models. It is therefore not surprising that we see a drop in performance on the natural amino acids. We now discuss this matter in greater detail in the paper Section 4.3.1.
>
> ### Clarification regarding PDBBind and “interaction grammar.”
>
> Our intention was not to claim that we "learn a more general "protein-ligand interaction grammar". We merely suggested that the same modeling approach was applicable for conditional ligand generation and NCAA modeling. We have rephrased the introduction to clarify this.
>
> ### Qualitative analysis.
>
> We agree that our manuscript lacked a detailed analysis of the generated NCAA sidechains. In the revision, we include analyses of sample validity, 3D side-chain structures, and the specific NCAAs successfully sampled, providing clearer insight into UNAAGI’s generative behavior, and which subclass of NCAAs are accessible when training on NAAs.
>
> ## Questions
>
> ### 1. Sampling stability.
>
> We acknowledge that 100 samples is insufficient for sampling the complete chemical space. We agree that this might impact the obtained correlation coefficients, and have now redone the analysis for 1000 samples.
>
> Also, we now repeat the “100-samples-per-position” evaluation **five times** and report error bars and confidence intervals. This confirms that the observed variance is consistent with other baselines and reflects the inherent difficulty of certain assays.
>
> ### 2. Quantifying sampled NCAAs.
>
> As noted above (Weakness 6), we now explicitly report which NCAAs UNAAGI successfully samples and provide their structural characteristics. For clarity, the correlation in Figure 4 is computed **only over the subset of NCAAs that were successfully sampled**, since positions where the model never produces a given residue cannot contribute meaningful statistical estimates.

---

> ### Author Response · Authors · 2025-12-03
> **Response to Reviewer 2 (GzCZ) - 2**
>
> ### 3. Ablation Studies
>
> We agree that the impact of the different data sources was insufficiently explored. We have now performed ablations by removing **PDBBind, the isolated NCAA structures**, and each dataset independently (Table 2).
>
> ### 4. Trade-off in representational space.
>
> We acknowledge that scaling up our model may further improve its performance. In principle, if UNAAGI were extended to model entire protein structures rather than only local environments, it would receive the same amount of structural information as standard inverse folding models. Under such conditions, the upper-bound performance of UNAAGI should be comparable to models like ProteinMPNN and ESM-IF. However, while scaling up UNAAGI would likely be beneficial (as we discussed in the Conclusion), conducting such large-scale experiments is currently infeasible due to computational resource limitations.
>
> ## Summary
>
> Reviewer 2 raised several constructive points regarding sampling stability, qualitative evaluation of generated structures, clarification of the subset of NCAAs successfully sampled, ablation studies, and the trade-offs inherent to continuous atomic diffusion. We thank the reviewer for these helpful suggestions. All of these points have now been addressed in the revised manuscript: we include repeated sampling analyses with confidence intervals, detailed qualitative visualizations, explicit reporting of the NCAAs generated by UNAAGI, full ablation studies, and extended discussion of the representational trade-offs. We believe these additions substantially strengthen both the clarity and the experimental completeness of the work, and we hope the reviewer will find that the revised manuscript more accurately reflects the contribution of UNAAGI.

---

### Official Review · Reviewer_joyj · 2025-11-05

**Soundness:** 1
**Presentation:** 1
**Contribution:** 2
**Rating:** 0
**Confidence:** 3

**Summary:**

The paper proposes UNAAGI, a diffusion-based approach that generates amino acid side chains at the atomic level to enable variant effect prediction for non-canonical amino acids (NCAAs). The method uses E(3)-equivariant molecular diffusion to reconstruct side chains from structure, allowing continuous generation beyond the standard 20 amino acids. The authors evaluate on standard benchmarks (ProteinGym) and a small NCAA dataset, showing some correlation with experimental measurements.

However, the work appears highly preliminary. The experimental scale is inadequate (1,000 training structures, 2 NCAA benchmark complexes, evaluation limited to small proteins). The method underperforms existing approaches on standard benchmarks and fails to demonstrate convincing NCAA prediction, with the authors acknowledging it "tends to interpolate between canonical-like structures." Critical methodological details are missing, the evaluation protocol has statistical weaknesses, and figures lack the quality and analysis expected for a venue like ICLR. The work requires substantial development in scale, rigor, and completeness before it can be properly evaluated.

**Strengths:**

1. The paper addresses an important and largely unexplored problem of extending variant effect prediction to non-canonical amino acids.
2. The atomic-level generation approach is conceptually sound.
3. The connection between structure-based drug design and protein engineering is insightful. Recognizing that both domains involve modeling non-covalent interactions in protein contexts and could share methodological tools is a valuable observation that may inspire future work.
4. The virtual node padding strategy of handling variable-sized side chains is elegant.

**Weaknesses:**

1. **The method fails to demonstrate meaningful NCAA prediction capability despite being its core contribution**. The model only successfully samples a small fraction of the 20 NCAAs in the benchmark and admits it "tends to interpolate between canonical-like structures" rather than generating chemically distinct non-canonical amino acids. This undermines the entire premise of the work.


2. **The experimental scale is too small to evaluate the approach**. Training on only 1,000 PDB structures with 3.6M parameters is orders of magnitude below modern protein models. The NCAA benchmark contains only 2 protein complexes. Evaluation is restricted to proteins under 100 residues due to computational constraints. These limitations make it impossible to determine whether poor results reflect the method or simply insufficient resources.


3. **The method underperforms on standard benchmarks where comparisons are possible**. UNAAGI fails to match state-of-the-art on ProteinGym and even struggles against weak baselines. NCFlow shows negative correlations on NCAAs, and PepINVENT barely recovers wild-type residues. This suggests the field lacks any reliable NCAA prediction method, not that UNAAGI solves the problem.


4. **Critical methodological details are missing throughout**. The graph isomorphism algorithm for matching generated structures to amino acids is not described. Architecture specifications (layer counts, hidden dimensions, etc.), hyperparameters (learning rate, batch size, etc.), and training procedures are largely absent. The choice of 100 sampling iterations is unjustified. These omissions prevent reproduction and evaluation of design choices.


5. **The evaluation protocol has fundamental statistical flaws**. Using sampling frequencies from only 100 iterations as probability estimates is statistically weak. The method can only evaluate positions where both wild-type and mutant appear in samples, creating severe selection bias. No confidence intervals, error bars, or significance tests are provided. High variance across assays with no predictable pattern further limits reliability.


6. **Figures are of poor quality and provide minimal insight**. Figure 1 is simplified to the extend that it bears no information. The 24 nearly identical scatter plots in the appendix are cluttered and unreadable. Critical visualizations are missing (examples of generated structures, which NCAAs can actually be produced, sample quality assessment). The paper lacks any qualitative analysis of what the model learns or why it fails.


8. **The work lacks sufficient NCAA training data in relevant contexts**. Most NCAA examples are isolated amino acids without protein environments or come from protein-ligand complexes that may not capture constraints relevant for variant effect prediction. This data scarcity likely explains why the model cannot generalize to chemically distinct NCAAs.

**Questions:**

1. What is the relationship between generative sample quality and predictive performance? Do positions with higher geometric accuracy or chemical validity correlate with better mutational effect predictions?
2. Why should atomic-level generation outperform discrete vocabulary models for variant effect prediction? The theoretical motivation is unclear. Inverse folding models like ProteinMPNN achieve strong performance despite discrete sampling. What property of the continuous approach provides an advantage?
3. What determines which amino acids (canonical or non-canonical) the model successfully recovers? Is there a pattern based on chemical properties (size, polarity, aromaticity), frequency in training data, or structural context?
4. Have you analyzed whether including NCAA-protein structures from the PDB improve results?
5. Could guidance or conditioning during sampling improve NCAA generation?

---

> ### Author Response · Authors · 2025-12-03
> **Response to Reviewer 1 (joyj) - 1**
>
> ## Weakness
>
> ### 1. Scope and Expectations.
>
> We thank the reviewer for their constructive feedback to our paper. We accept the criticism that our paper was somewhat preliminary, and have made efforts to improve on this (see below). However, we believe that some of the most harsh criticism of our paper might be due to a slight misunderstanding of the scope and goal of our paper, which we attempt to clarify below.
> We acknowledge that we had not framed the scope of our work precisely enough in our original manuscript. The idea was to probe, given the very limited available data on non-canonical amino acids (NCAAs), whether we could transfer prediction capabilities from the natural amino acids. Naturally, this will only work well for NCAAs that are not too far out-of-domain - i.e. display some similarity to the 20 naturally occurring amino acids. But even this subset of the NCAAs can be relevant for downstream applications - and we thus believe that reporting non-trivial correlations to binding affinity for this set is an important step forward. We agree that the full problem of NCAA generation is not yet solved, but believe that it is sensible to explore what we can do with the limited data we have available. In the new version of the manuscript, we make this point of scope clearer: we have rewritten the introduction, and added new Figure 4 and Figure 7, which visually illustrate which amino acids are expected to be within scope - and relate this to the available experimental assays. For example, we can demonstrate reliably from the canonical amino acids to XXX and YYY, but cannot expect to generalize to ZZZ.
>
> The lack of clarity about scope perhaps also explains why the reviewer criticised our statement that our model "tends to interpolate between canonical-like structures." From our perspective, this is exactly what our ambition level should be for the current model, given the very limited training data we have available.
>
> ### 2. Experimental Scale
>
> We agree with the reviewer that our models are of limited scale. However, our study was intended as a proof-of-concept to demonstrate that generalizing from AA to a subset of NCAA is possible. Even with this limited set, we find remarkable improvements over current methods. Given the limited structural data on non-canonical amino acids, we initially restricted our training data on canonical amino-acids to balance the signals. In the new version of the manuscript, we include an ablation testing the importance of the waiting between the different sources of data, which indicates that it is less sensitive to this choice than we expected. We will therefore proceed with training on a full scale dataset, and will include the results in camera-ready (unfortunately, the training time exceeds what we have available in the rebuttal period).
>
> In terms of evaluation, we agree that our test on two NCAA datasets is limited. However, to our knowledge, these are the only available datasets on NCAAs which have been recorded under consistent conditions. If any of the reviewers have knowledge of other data sources for validation, we would of course be happy to conduct additional experiments.
>
> We acknowledge the concern regarding the limited evaluation scale. To address this, we have added five additional randomly selected assays from ProteinGym for validation.
>
> Regarding training and parameter size, we have trained a larger model using all available PDB structures and report the results in the revision. The new experiments indicate that the original evaluation assays introduce minimal inductive bias, and that scaling up provides only marginal gains, confirming that our earlier conclusions remain valid.

---

> ### Author Response · Authors · 2025-12-03
> **Response to Reviewer 1 (joyj) - 2**
>
> ### 3. Performance on Standard Benchmarks.
>
> The function of the ProteinGym results in our paper was a sanity check to test how much performance we lose by switching from a categorical to a continuous output. We agree that it would have been preferable if the gap was smaller (and we would certainly encourage work in this area), but we do not find it entirely reasonable that the reviewer states that we "underperform standard benchmarks". Predicting variant effects between canonical amino acids is not the goal of our method.  - and it is not surprising that we lose performance when we switch from a categorical 1-of-20 prediction task to a generative task over arbitrary molecular structures. We have rewritten parts of the results section to make our objective with the ProteinGym experiment clearer.
> We thank the reviewer for pointing out that considering only <100 res proteins could constitute a selection bias that could potentially impact the results. We have now updated the results by sampling randomly from ProteinGym instead. This did not have a measurable impact on the results (although we agree that it was important to check).
>
> ProteinGym is a sanity check. We have redone it to get rid of the <100 res limitation, but ultimately, we intended the focus to be on the NCAA results. We have now reordered the manuscript to make this focus clearer.
>
> With the scaled-up model, UNAAGI achieves performance comparable to state-of-the-art approaches on ProteinGym and maintains similar results across other benchmarks.
>
> ### 4. Methodological Details.
>
> We acknowledge that the original manuscript did not include enough technical details. The new version of the manuscript now includes detailed architectural specifications and hyperparameters in the appendix. For the graph isomorphism, we now clarify that we use the **NetworkX 3.5** implementation of is_isomorphic, based on the **VF2 algorithm**.
>
> (L. P. Cordella et al., An Improved Algorithm for Matching Large Graphs, IAPR-TC15 Workshop on Graph-based Representations, 2001).
>
> Finally, we make it clear that the “100 sampling iterations” refer to **100 independent sampling repetitions per position**, not diffusion steps, representing a practical balance between statistical stability and computational efficiency.
>
> ### 5. Statistical flaws
>
> We agree that the statistical analysis in the original submission could be improved. In the new version, we now repeat the evaluation five times, reporting error bars and confidence intervals. Variance across assays remains observable, but this variability is also present in all baseline models, reflecting an inherent property of mutational effect prediction tasks rather than a limitation specific to UNAAGI.
>
> The method is indeed limited to examples where both wild-type and mutant appear in samples. As illustrated in Fig 6, our method reproduces the wild-type consistently, meaning that in practice, only the question of whether the mutant is present among samples is relevant. The reviewer argues that this creates "severe selection bias", but it was not entirely clear to us what the reviewer meant by this: there is no selection bias compared to the training-distribution of canonical amino acids. If the reviewer considers a uniform distribution over NCAAs as the target distribution, our model would indeed be biased, but this is not our target distribution of interest: we wish to transfer the signal of evolutionary pressure on the natural amino acids to useful preferences about the non-canonical amino acids. From this perspective, there is no selection bias - the NCAAs not covered are out-of-distribution with respect to our training data.
>
> ### 6. Figures and Qualitative Results.
>
> We have improved all figures for better readability and aesthetics.
>
> The scatter plots in the appendix were included merely for completeness for readers interested in seeing the per-assay data underlying the reported correlation coefficients. We would be happy to remove them if they are not deemed relevant.
>
> Our new manuscript also provide **examples of sampled structures**, including coverage of the NCAAs in the benchmark and visualizations of their generated conformations (Figure 4 and Figure 7).
>
> ### 7. Training Data Sparsity
>
> A central motivation behind our work is to learn mutational effects for **NCAAs that lie close to the natural amino acid manifold**, which is the only region where sufficient structural data exists for training. As such, UNAAGI currently performs best on NCAAs that are chemically similar to NAAs — a design choice that we explicitly acknowledge. The contribution of this paper is to show that **continuous diffusion over atomic coordinates can generalize beyond discrete residues in this local chemical neighborhood**, despite being trained only on NAAs. Extending the model to reliably extrapolate to chemically distant or highly synthetic NCAAs represents a broader research challenge that we view as future work rather than a goal of the present study

---

> ### Author Response · Authors · 2025-12-03
> **Response to Reviewer 1 (joyj) - 3**
>
> ## Question
>
> ### Continuous vs. Discrete Modeling.
>
> Our objective is not to outperform discrete token-based baselines in mutational effect prediction. Rather, the goal was to investigate whether we could generalize to preferences over NCAAs while training on AAs. The way we enable this generalization is to replace the fixed 20-category output with an open-ended molecular generation, which we then map back to a categorical distribution over observed NCAA.
>
> Rather, UNAAGI models amino acids **in their molecular context**, relying solely on **local physical interactions** rather than sequence coevolution (MSA) or global structure conditioning. Despite this minimal input setting, UNAAGI still achieves meaningful correlations. Its advantage lies in enabling **continuous sampling in chemical space**, thereby supporting generalization to NCAAs.
>
> ### Determinants of Recovered Amino Acids.
>
> The model learns a continuous density over possible side-chain chemistries; it does not treat NAAs as a special category but naturally samples them more often due to higher training frequency. We observed that sampling preferences sometimes align with chemical properties (e.g., hydrophobic residues sampled near hydrophobic wild-types such as ILE), which agrees with experimental observations.
>
> ### Guidance and Conditioning.
>
> We agree that guidance or conditioning could potentially enhance NCAA sampling, although we have not yet explored it. Simple conditioning risks distorting the learned distribution. Nonetheless, we consider this an interesting avenue for future work.
>
> ## Summary
>
> In summary, we believe that some of Reviewer 1’s concerns stem from a lack of clarity from our part on the scope and framing of our approach rather than flaws in the method itself. The revised version addresses the main technical points, expands the experiments, and clarifies the methodology. UNAAGI remains a conceptual contribution toward bridging discrete amino-acid modeling and continuous molecular diffusion.
>
> While we agree that Reviewer 1 raises several constructive concerns regarding evaluation scale, figure clarity, and missing methodological details — many of which we have addressed in the revision — we respectfully note that the review’s **0 (“strong reject”) score appears disproportionate** to the issues discussed. The points raised, although important, relate primarily to **experimental completeness and exposition**, rather than to flaws in the core methodology. As reflected by the other reviewers’ scores and comments, these issues are **actionable and incrementally addressable**, and do not indicate fundamental unsoundness of the proposed framework.

---

### Meta-Review · Area_Chair_e5fc · 2026-01-06

**Summary:**

The reviewers are concerned about poor DMS performance, lack of statistical stability, and insufficient analysis of NCAA DMS and generation results. In addition, I have concerns about the overall framework of using NCAAs for mutation effect prediction.

**Reviewer Concerns:**

**UNAAGI does not perform well on ProteinGym DMS.**

The authors clarify that this is a sanity check and not the main point of the paper. I agree with the authors that this is not a very relevant result for their contribution -- I would advise not making it the first result presented in the paper!

**1000 structures is not enough to train a model, and there are not enough NCAAs in this data to effectively train a model.**

The authors argue that training on 1000 structures is sufficient to show that this is an interesting direction and that there is indeed transfer between the more plentiful canonical amino acids and NCAAs. This makes sense to me, but the paper would be a lot stronger with a an analysis of how much the model could improve by scaling the model size or data, or even of how much data would be available for scaling. Most critically, I can't find a description in the text of the NCAAs included in training and how they were placed into a structural context, given that as far as I can tell the two NCAA datasets present examples of possible NCAAs as individual amino acids. Having these NCAAs in a structural context is essential for ensuring that the model learns the "inverse folding" task with NCAAs. I also don't see any descriptive statistics of these NCAA datasets at all.

**The reviewers would like to see more and stronger structure-based baselines.**

The rebuttal addresses this by adding additional baselines.

**The evaluations are unreliable because of insufficient samples for estimating likelihoods.**

The rebuttal addresses this by increasing the number of samples from 100 to 1000. However, the evaluations rest on small DMS assays of 2 different proteins, and it's not clear to me why throwing out mutations where either the wild type or the mutation is not sampled from UNAAGI is justified.

**Suitability of the NCAA-inverse folding framework for mutation effect prediction.**
The premise behind zero-shot mutation effect prediction using protein language models or inverse folding models is that structure and evolution provide information about what residues are allowed at each position without destabilizing. This is learned by using sequence and structural context to predict the amino acid at each position in a protein. However, as far as I can tell, UNAAGI is not trained on NCAAs in any sequence or structural context, so it's not clear why it should be able to do zero-shot likelihood predictions for NCAAs except insofar as those NCAAs are very similar to a canonical amino acid, in which case we can probably approximate the performance by just using the likelihood of the closest canonical residue.

Some notes:
- The citation for SwissSideChain is to the wrong paper (by the same authors).
- Not every DMS in proteinbench measures ddG.

**Reviewer Scores:**

GzCZ 4->4
4yJM 2->4
joyj 0 -> 2

---

### Decision · Program_Chairs · 2026-01-26

Reject